# Transition from light diffusion to localization in three-dimensional amorphous dielectric networks near the band edge

Jakub Haberko [1,3], Luis S. Froufe-Pérez [2,3] & Frank Scheffold [2✉]

Localization of light is the photon analog of electron localization in disordered lattices, for whose discovery Anderson received the Nobel prize in 1977. The question about its existence in open three-dimensional materials has eluded an experimental and full theoretical verification for decades. Here we study numerically electromagnetic vector wave transmittance through realistic digital representations of hyperuniform dielectric networks, a new class of highly correlated but disordered photonic band gap materials. We identify the evanescent decay of the transmitted power in the gap and diffusive transport far from the gap. Near the gap, we find that transport sets off diffusive but, with increasing slab thickness, crosses over gradually to a faster decay, signaling localization. We show that we can describe the transition to localization at the mobility edge using the self-consistent theory of localization based on the concept of a position-dependent diffusion coefficient.

---

[1] Faculty of Physics and Applied Computer Science, AGH University of Science and Technology, Al. Mickiewicza 30, Krakow 30-059, Poland. [2] Department of Physics, University of Fribourg, Fribourg 1700, Switzerland. [3] These authors contributed equally: Jakub Haberko, Luis S. Froufe-Pérez. ✉email: Frank.Scheffold@unifr.ch

Bandgap formation and strong Anderson localization (SAL) of classical waves are both considered general wave phenomena where the mechanism leading to the exponential attenuation of wave transport can be understood in terms of interference of scattered waves. In a periodically repeating environment, scattering of waves from Bragg planes can be associated with the opening of a photonic bandgap (PBG)[1–3].

The SAL mechanism is usually explained by the constructive interference of multiply scattered waves propagating along time-reversed loops, which increases the return probability and eventually leads to a breakdown of wave diffusion[4–6]. In contrast to PBG formation, the transition to SAL in disordered media strongly depends on dimensionality. In one and two dimensions, there are no truly extended states, and waves in disordered media are always localized for sufficiently large systems sizes[7–11]. Only in three dimensions, wave localization shows a phase transition, and localized states appear below the 'mobility edge'. The threshold for localization can be estimated by the Ioffe-Regel criterion $k \cdot \ell \sim 1$, where $\ell$ denotes the transport mean free path and $k$ the wavenumber[12–14]. The Anderson transition in three dimensions is a fascinating phenomenon that until now has eluded a full theoretical and experimental understanding. It's relevance is not restricted to transport of photons or electrons, but it can also be applied in acoustics, or any kind of coherent wave propagation[15]. Moreover, a full understanding and control over the wave fields in complex media offer a plethora of opportunities for applications in imaging, sensing, and photonics[6]. However, an experimental observation of SAL for electromagnetic vector waves in three-dimensional systems has not yet been achieved, even though several claims to its existence were made[16–18] but soon after were put in question[19,20] and later refuted[21]. At about the same time it was found theoretically that SAL of electromagnetic vector waves is absent in random ensembles of point scatterers, irrespective of their scattering strength[22]. Thus, in a 2016 perspective article, Skipetrov and Page[21] declare a "Red light for Anderson localization" of light and Maret et al.[23] ask "Can 3D light localization be reached in 'white paint'" at all?

The advent of amorphous PBG materials over the last decade[24–29] has opened a new pathway toward the design of strongly photonic dielectric materials. It has also prompted fundamental questions concerning the relationship between the Anderson localization transition and the transition to a full bandgap. Recently, we proposed a transport phase diagram for two-dimensional hyperuniform disordered PBG materials with a SAL regime near the PBG[30] and conjectured that these findings could be generalized to three dimensions[24–27,30,31]. Early pioneering work by John suggested the possibility of finding SAL in disordered crystalline structures near the bandgap[2,32,33], an idea supported more recently by the numerical studies of Conti and Fratalocci[33]. Imagawa et al. reported an increase of the inverse participation ratio near the bandgap of an amorphous diamond structure, which might indicate the presence of localized states[34]. This previous work provides a rationale for the existence of a SAL regime in the vicinity of a bandgap of an amorphous photonic material, which we are going to investigate in our work.

To this end, we study the transport properties of electromagnetic vector waves in realistic digital representations of three-dimensional hyperuniform silicon networks numerically. We find evidence for the anomalous light transport near the band edge, signaling the onset of Anderson localization at a mobility edge and a broad frequency window where light is localized before the bandgap fully develops.

## Results
### Strong Anderson localization of waves
The discussion of what defines SAL is very detailed and rich, and for a comprehensive review, we refer to the literature[5,13,21,32,35–38]. A working definition for finite-sized systems, proposed by Cherroret and Skipetrov, is that 'SAL is an interference wave phenomenon in a medium of a finite size that would give rise to truly localized states if the medium were extended to infinity'[38]. The transition to SAL in three dimensions is usually described in the framework of the self-consistent (SC) theory[39]. It treats localization by introducing a position $\overrightarrow{\mathbf{r}}$ dependent wave diffusion coefficient $D(\overrightarrow{\mathbf{r}})$, which decays to zero deep inside the medium[5,38], due to an increased return probability of multiple scattering paths. At the mobility edge, for a semi-infinite medium, one finds the simple algebraic forms $D(z) = \frac{D(0)}{1+z/\xi_c}$ and in the localized regime $D(z) = D(0) \exp(-2z/\xi)$ where $z$ denotes the distance from the surface. The localization length $\xi$ becomes finite at the critical point while $D(z) < D_B$ already when approaching the mobility edge, where $D_B = c\ell/3$ denotes the standard Boltzmann diffusion constant. $D(0)$ continues to drop gradually as the localization threshold is crossed[40]. For slabs of finite thickness $L$, the transmittance $T(L)$ shows two distinct regimes: initially, it decays diffusive as $\sim L^{-1}$ which is followed by an exponential decay $\sim e^{-L/\xi}$. Precisely at the transition, SC theory predicts a critical power-law scaling $T \sim L^{-2}$ instead of the exponential decay.

We point out that the (SC) theory is an approximate theory, and different variants have been proposed in the literature[5,38,41]. As discussed in ref. 38, in its original form, it is strictly valid only for $k \cdot \ell \gg 1$. The fact it can describe certain phenomena at the mobility edge and in the localized regime is somewhat fortuitous and not fully understood. However, progress toward a better microscopic understanding has been reported recently[38,42]. Moreover, almost all theoretical and numerical studies were carried out assuming scalar waves and point scatterers randomly distributed in space, i.e., using white-noise Gaussian statistics[14]. It is not self-evident that the conventional SC theory[40], can explain the transport of vector waves in spatially correlated, densely filled dielectric materials with a bandgap. First of all, the transport $\ell$ and the scattering mean free path $\ell_s$ are not the same for scatterers of size $\sim\lambda$, and the wave is propagating in some effective medium with a wavenumber $k_{eff}$[11,30]. If and how this affects the predictions by SC theory is currently not known. As a consequence of the approximate nature of the SC theory, in particular, when applied to realistic representations of dielectric materials, we must assume that there is no perfect one-to-one relationship between the macroscopic transport properties, such as the localization length $\xi$ and microscopic quantities like $k = 2\pi/\lambda$ and $\ell$. We, therefore, denote with $(kl)$ the localization parameter, which we assume is similar and proportional but not necessarily identical to $k \cdot \ell$. For simplicity, we also assume $(kl)_c \equiv 1$[14], and we have tested that using slightly different values for the localization threshold does not significantly affect our findings.

In summary, in our study the localization parameter $(kl)$ sets the macroscopic properties in the SC theory, such as $\xi/\ell = 6(kl)^2/(1-(kl)^4)$ for $(kl) < 1$ and $D(0)$, with $D(0) = D_B(1-(kl)^2)$ for $(kl) \gg 1$[40]. The relationship to the microscopic parameter $\ell$ is established via $(kl) \propto k \cdot \ell$ modulus a prefactor of order one that also takes account of the uncertainty with respect to $\ell/\ell_s$ and the effective wavenumber $k_{eff} \gtrsim k$. We note that the predictions by SC theory are only meaningful in the limit $\xi/\ell \gg 1$. For example $\xi/\ell > 5$ for $(kl) \in [\frac{3}{4}, 1)$.

**Transport in a photonic bandgap.** A photonic crystal with a full PBG displays a vanishing density of states (DOS) in the gap. Transport through a finite-sized slab is due to tunneling, characterized by an exponential attenuation with a decay length $L_B$, called the 'Bragg length,' typically on the order of one unit cell, in high refractive index photonic crystals[43,44]. For frequencies

outside the gap, or for specific directions in the presence of an incomplete gap, photonic crystals are transparent, before the onset of diffraction[43]. Perfect crystals show neither photon diffusion nor localization. In his celebrated 1987 paper, John suggested that three-dimensional photonic crystal lattices with moderate disorder may exhibit strong localization of photons[2], an idea supported more recently by numerical studies[33]. Amorphous PBG materials are disordered but the spatial distribution of dielectric material is highly correlated, which can also lead to the opening of a full PBG. The genuine disorder, however, implies that these materials display strong scattering for frequencies outside the gap, or if the gap is incomplete for all frequencies, with the notable exception of transparency in stealthy hyperuniform materials in the long-wavelength limit[45,46], which we do not address here. Strong scattering outside the gap opens the possibility for the existence of SAL transport regimes, even in the absence of defect states[30].

**Optical transport simulations and density of states (DOS).** We study the transport of waves in three-dimensional hyperuniform silicon photonic network structures, refractive index $n = 3.6$, derived from the center positions of an assembly of 10,000 randomly close-packed spheres, diameter $a$, as described earlier[47,48]. The design protocol consists of mapping the seed pattern into tetrahedrons by performing a Delaunay tessellation. Then, the centers of mass of the tetrahedrons are connected, resulting in a

tetravalent network structure of interconnected rods with the desired structural properties. The diameter of the rods sets the space-filling fraction $\phi$, Fig. 1a, c. The length $a$ is the typical short-range structural length scale of the network, which for a crystal would be the lattice constant. The seed point pattern is hyperuniform, but not stealthy, meaning that the isotropic structure factor vanishes asymptotically in the limit $S(k) \to 0$ for $k \to 0$ where $k = 2\pi/\lambda$ denotes the wavenumber in vacuum. Practically identical network structures have been considered by Liew et al. in a study of the optical DOS[26] (see also Supplementary Figs. 2 and 3). They report a substantial depletion of the DOS, by more than two orders of magnitude, over a significant range of frequencies, indicating the presence of a bandgap for different values of $\phi$. In the present study, we consider networks with a filling fraction of $\phi = 0.28$, shown to display the most pronounced photonic properties in their study[26].

For the optical transport simulations, we apply the finite differences time-domain (FDTD) approach, implemented by the MIT Electromagnetic Equation Propagation (MEEP)[49]. It is considered to be one of the most potent simulation techniques to study electromagnetic wave transport. In a single MEEP-simulation run, a broadband pulse of linearly polarized light, with an electric field vector parallel to one of the sides of the simulation box, is incident on the sample, as illustrated in Fig. 1a. We obtain the full spectrally resolved information about the optical transmittance $T(a/\lambda, L)$, Fig. 1b. We present all spectra in terms of the reduced frequency $\nu' := a/\lambda = \nu a/c$ where $\lambda$, $\nu$, and $c$ denote the wavelength, frequency and speed of light

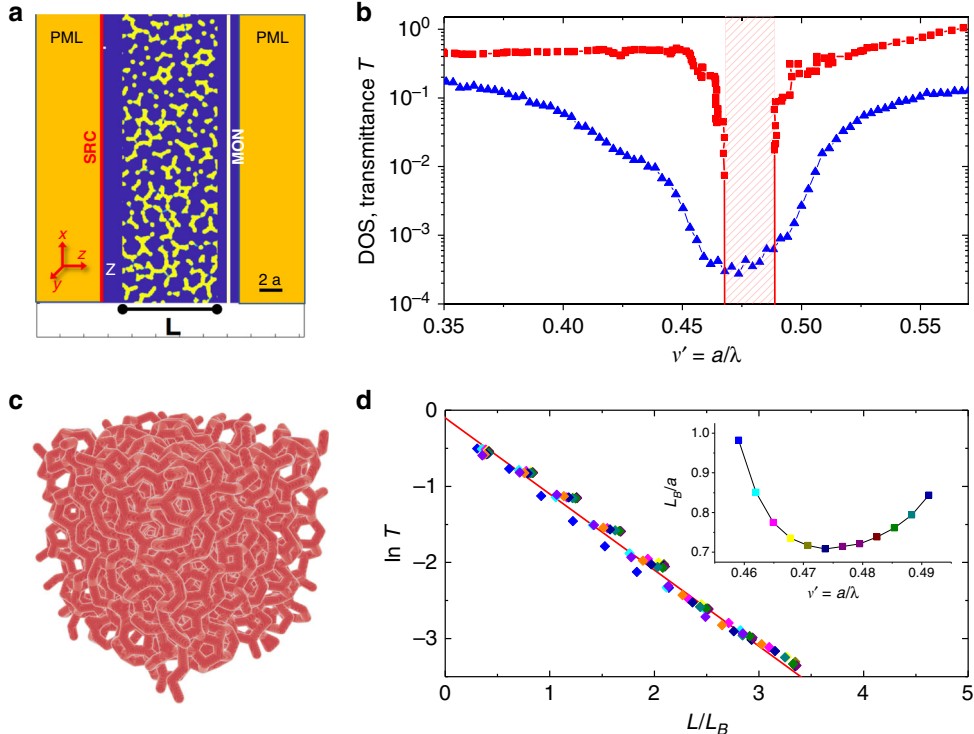

**Fig. 1 Numerical simulations of optical transport properties. a** Cross section of a three-dimensional FDTD (MEEP) simulation box containing a hyperuniform silicon ($n = 3.6$) network structure, thickness $L = 6a$. A light wave, linearly polarized along the $x$-axis, is launched on the leftside and propagates along the $z$-axis. The photonic network structure is terminated with perfectly matched layers (PML) at both sides of the box along the propagation axis. PMLs act as absorbers. The source (SRC) and detector (MON) are placed at a distance approximately $\sim a$ from the sample, which is held in vacuum. Periodic boundary conditions are applied along $x$ and $y$ directions. **b** Triangles: transmittance spectrum $T(a/\lambda)$ for a slab of thickness $L = 18a$ for a filling fraction $\phi = 0.28$. The optical transport data is compared to numerical calculations of the density of states (DOS) (squares). The bandgap-center frequency is $\nu'_{\text{Gap}} = a/\lambda_{\text{Gap}} = 0.478$ and the width $\Delta\nu$ is indicated by the shaded area. **c** Three-dimensional rendering of a hyperuniform network structure, edge length $6a$ and filling fraction $\phi = 0.28$. The size of the structure used in the simulation is $18a \times 18a \times L$ with $L \le 18a$, which is repeated periodically in ($x$, $y$) direction to construct the slab geometry. **d** In the gap the transmittance decays exponentially and $\ln T$ collapses on a master curve when plotted in reduced units $L/L_B$. $L_B \le a$ denotes the Bragg length and it is found to be smallest near the gap-center frequency $\nu'_{\text{Gap}} = 0.478$, see also Supplementary Fig. 1.

| **Table 1 Values of $L_i/a$ used used in all fitting procedures.** | |
| --- | --- |
| $i$ | $L_i/a$ |
| 1, 2, 3, 4, 5, 6, 7, 8, 9, 10, 11, 12, 13, 14, 15 | 2.7, 3.0, 3.6, 4.8, 6.0, 7.2, 8.4, 9.6, 10.8, 12.0, 13.2, 14.4., 15.6, 16.8, 18.0 |

in vacuum[5]. We obtain results for slabs of thickness $L = 0.3a{-}18a$ and average the simulations results over 6 (thick slabs) to 15 (thin slabs) independent configurations of the network structure. We note that the networks are structurally isotropic[47] and therefore, all incident polarization states on average lead to the same results.

To obtain the normalized photonic DOS we use the supercell method[3], implemented in the open-source code MIT Photonic Bands (MPB)[50], Fig. 1b. Due to computational limitations, we have generated equivalent, but smaller seed patterns with periodic boundary conditions applied. To this end, we are using a packing algorithm developed by Skoge et al.[51].

**Fitting procedure**. The transmittance spectrum $T(\nu', L)$ depends on a number of a priori unknown parameters. Extracting all these parameters from a global fit to SC theory is not stable and prone to overfitting. To overcome this problem, we first determine parameters that do not scale with $\nu'$, such that eventually, $T(\nu', L)$ predicted by SC theory, depends only on the one parameter $(kl)$. Once this is achieved, the predictions by SC theory and diffusion theory can be compared without fitting bias.

We test the presence of three transport regimes: (1) evanescent transport $\ln[T(\nu', L)] \sim -L/L_B$ in a PBG with a Bragg length $L_B \sim \ell$. (2) Diffusive transport with $D(z) \approx$ const. and (3) SAL, as described by the SC theory with $(kl) \lesssim 1$ and $\xi \gg \ell$. We proceed as follows. (1) We start by looking for the signature of an evanescent decay in the gap. (2) Next, we consider frequency intervals far below and above the gap, where the predictions for $T(L)$ from classical diffusion theory are sufficient to describe the data. This is the case whenever $L \gg \ell$ and $D(z) \approx$ const. From this fit we extract the angle averaged reflection coefficient $R$. (3) Next we identify the position(s) of the mobility edge(s) $\nu'_c$, where $(kl) = (kl)_c \equiv 1$, by fitting the data with SC theory treating both $(kl)$ and and $\ell$ as adjustable parameters. From this fit we find the anchor points $\nu'_c$ where $(kl) = 1$. Together with $k = 2\pi/\lambda = 2\pi\nu'/a$, this provides us with the proportionality between $k \cdot \ell$ and $(kl)$. (4) Now, that we have fixed all other parameters, we will attempt to describe the entire data set for $T(\nu', L)$ by SC theory with only one adjustable parameter $(kl)[\nu']$.

**Density of states and tunneling through the gap**. Our numerical calculations of the DOS reveal a full bandgap for frequencies in the interval $\nu' \in [0.47, 0.49]$, Fig. 1b (see also Supplementary Methods). Our results for the gap-center position and the width of the gap are in good agreement with an earlier study by Liew. et al. using a different method but applied to a practically identical system at $\phi = 0.28$ volume filling fraction[26], see also Supplementary Fig. 2. Indeed, for $\nu' \in [0.47, 0.49]$, where the DOS is zero, we find that $T(L)$ decays exponentially with $L_B < a$, Fig. 1d. The decay length rises toward the band edges and is smallest around the center frequency $\nu'_{Gap}$. This observation is consistent with tunneling of evanescent waves through the whole sample of thickness $L$[3,44]. We find that the evanescent regime appears to extend over a slightly broader range of frequencies compared to the bandgap. We will address this point again at the end of this section.

**Transmittance in the multiple scattering regime**. Standard diffusion theory describes transmission through samples whose

thickness $L$ is much larger than the mean free path $\ell$ and thus $T \sim \ell/L \ll 1$. Since the size of our simulation box is limited to $L \leq 18a$, we also have to include data for slabs with thicknesses on the order of a few $\ell$, in particular far from the gap where the transmittance $T$ can be closer to one. To be able to describe the transition from ballistic to diffusive transport as well as to SAL we follow an approach developed by Durian and coworkers. Their theory, which is based on the telegrapher equation, takes into account ballistic transmission, lower order scattering as well as diffusive multiple scattering of light. Their theory accurately predicts $T_{L\to0} = 1$ for thin samples, while diffusion theory fails in this limit. In the methods section we explain how to consistently merge Durian's approach[52] with the SC theory of localization and we obtain:

$$T = \frac{z_0 + D_B/D(0)}{2z_0 + \tilde{L}/\ell}\left(1 - e^{-L/\ell}\right) - \frac{\tilde{L}/\ell}{2z_0 + \tilde{L}/\ell}e^{-L/\ell} + \frac{\eta(L/\ell)}{2z_0 + \tilde{L}/\ell} + e^{-L/\ell},$$

$$\tag{1}$$

where $\tilde{L}$, $D(0)$ and $\eta$ depend on $(L, \ell)$ as well as the localization parameter $(kl)$. $z_0$ denotes the extrapolation length in diffusion theory. We note that merging Durian's theory with SC theory is unproblematic. The improvements of the former affect thin slabs $L \sim \ell$ while the latter only affects thick slabs $L > \xi \gg \ell$. In essence, Eq. (1) provides a proper interpolation scheme between the two limiting cases. In the absence of SAL, for $(kl) \to \infty$, we recover Durian's results for $T(L/\ell, z_0)$ with $\tilde{L} \equiv L$, $D_B/D(0) = 1$ and $\eta(L/\ell) = 0$. Then, for $L \gg \ell$, Eq. (1) reduces to the common expression for diffuse transport $T(L \gg \ell, z_0) = \frac{1+z_0}{2z_0 + L/\ell}$.

When deriving Eq. (1) we did not distinguish between the scattering and the transport mean free path and only use one $\ell$ for the mean free path. By considering an extended version of the model, taking into account the scattering anisotropy parameter $g$, as described in ref. [52], we have verified that these simplifications do not adversely influence the quality of our fit. Moreover, we neglect specular reflections at the interface of the order of a few percent at most.

In diffusion theory, the extrapolation length $z_0$ is linked to the angular averaged specular reflectivity $R$ via $z_0 = \frac{2}{3}\frac{1+R}{1-R}$. We determine $R$ directly from a fit to the data using a least-squares fitting procedure. Since the transmittance varies by several orders of magnitude and we are interested in the behavior in different regimes, we choose the natural logarithm of the transmittance as the function to fit. For a given frequency we define:

$$S \equiv \sum_{i=1}^{N}\left[\ln\left(T_{FDTD}(L_i)\right) - \ln\left(T(L_i)\right)\right]^2,\tag{2}$$

as the sum of squares to be minimized. $T(L_i)$ is defined according to Eq. (1). The index $i$ runs over the $N = 15$ data points from $L/a = 2.7$ to $L/a = 18$, see Table 1. We note that $\ell$ sets the optical thickness in terms of the characteristic length $a$ in our system: $L/a = L/\ell \times \ell/a$. We first fit Eq. (1) to the numerical data, assuming the absence of localization or $(kl) \to \infty$. Our analysis covers the entire frequency range considered, $\nu' \in [0.3, 0.6]$ treating both $R$ and $\ell$ as adjustable parameters. Figure 2a shows the frequency dependence of $S$ and the values of $R(\nu')$ we obtain. Far from the gap-center frequency $|\nu' - \nu'_{Gap}| \gtrsim 0.1$ we find small $S$ values signaling excellent agreement between diffusing theory and data, as shown in Fig. 2b. The good fit suggests a classical transport regime controlled by ballistic transmission, low order scattering, which eventually evolves to become diffusive for $L \gg \ell$. In the same frequency range we find $R = 0.66 \pm 0.05$ to be approximately constant, corresponding to $z_0 \simeq 3.25 \pm 0.5$. We repeat the fit keeping $R = 0.66$ fixed and the goodness of the fit is

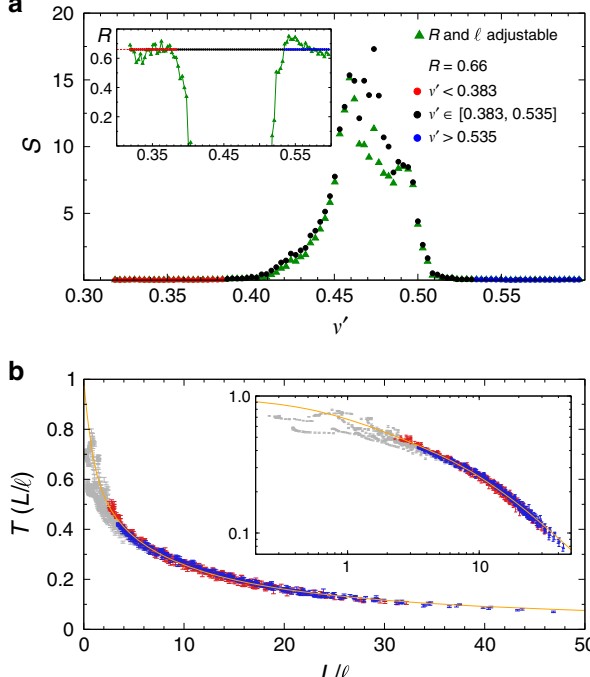

**Fig. 2 Comparison to diffusion theory. a** Error of the least squares of the fit of Eq. (1), in the absence of SAL (($kl$) → ∞), to the data with $R$ and $\ell$ as adjustable parameters (triangles). In the frequency range marked with red and blue full circles, we find $R = 0.66 \pm 0.05$ or $z_0 \simeq 3.25 \pm 0.5$, both in the higher and lower frequency branch (inset). Full circles: results from least-squares fitting when setting $R = 0.66$ constant. **b** Far from the gap, for $\nu' \neq [0.383, 0.535]$, transport is diffusive and all data for $T(L/\ell)$ can be collapsed on a master curve given by Eq. (1) (yellow line) with $\tilde{L} = L$ and $z_0 = 3.25(R = 0.66)$. Red (blue) full circles refer to the same data at frequencies lower (higher) than $\nu'_{\text{Gap}}$ shown in (**a**). Data for $L < 2.7a$, not taken into account for the fitting procedure, are marked with gray symbols.

the same, Fig. 2a. For comparison, we have calculated the internal reflection coefficient from an effective medium interface $n_{\text{eff}} = 1.42$ (for $\phi = 0.28$) to vacuum and find a very similar value $R \simeq 0.73$, which demonstrates the consistency of our fitting procedure (for details see Supplementary Methods and Supplementary Fig. 4). We note that around $\nu' = a/\lambda_{\text{Gap}} \sim 0.45$ the fit is very poor and the fitted values of $\ell$ and $R$ become meaningless.

**Self-consistent theory of localization.** To assess the breakdown of wave diffusion near the band edge, we compare our numerical data to the SC theory of localization. For slabs of finite thickness $L$ we can calculate $D(z)$ for a given value of ($kl$)[40]. To illustrate the dependence of the diffusion coefficient on $z$, in Fig. 3 we plot the $D$ as a function of $z/L$ for different values of $L/\ell$ and ($kl$). As expected, for ($kl$) greater than one the diffusion coefficient shows a weak dependence on both the total size of the system and the position. On the contrary, deep in the localization regime, $D(z)$ decays exponentially away from the boundaries. Exactly at the localization transition for ($kl$) = 1, the diffusion coefficient is strongly reduced in the center of the slab but the decay remains nonexponential.

We integrate $D(z)$ to obtain $\tilde{L} = \int_0^L D_B/D(z)\,dz$ and the function $\eta(L/\ell)$. For ($kl$) values larger than one, SC theory gradually approaches the prediction by diffusion theory and the quality of the fit becomes insensitive to the choice of ($kl$). As shown in Fig. 4a, b the $S$ values of both diffusion theory and SC theory become comparable for $\nu' < 0.4$ and $\nu' > 0.51$, signaling a

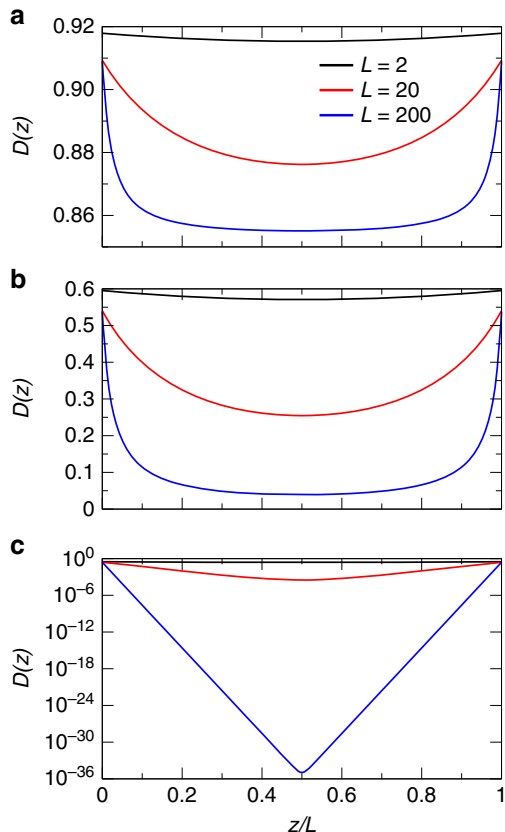

**Fig. 3 Position dependence of the diffusion coefficient in the SAL regime.** $D(z/L)$ for ($kl$)$_c$ = 1, $L/\ell = 2$ (black), $L/\ell = 20$ (red) and $L/\ell = 200$ (blue). **a** $kl = 2.6$; **b** $kl = 1$; and **c** $k\ell = 0.6$.

diffusive transport regime. When approaching the gap from lower (or higher) frequencies, the fit with SC theory, however, leads to substantially smaller $S$ values, indicating localization. From the two-parameter fit we find a lower frequency mobility edge ($kl$) = 1 at $\nu'_{c,l} = 0.412$ ($\ell_{c,l}/a = 0.513$) and a higher frequency mobility edge at $\nu'_{c,h} = 0.506$ ($\ell_{c,h}/a = 0.242$), Fig. 4c, d. Using these anchor values for the mobility edge, ($kl$) and $\ell$ are linked via the relation ($kl$) = $\ell/\ell_c \times \nu'/\nu'_c$. With ($kl$) ≡ 1 at the mobility edge, we find $k \cdot \ell = 1.33$ for $\nu'_{c,l} = 0.412$ and $k \cdot \ell = 0.77$ for $\nu'_{c,h} = 0.506$. We note that we use these two separate proportionality constants (1.33, 0.77) for the comparison between theory and numerical data in the higher and lower frequency branch. In all cases, we perform a least-squares fit to ln T according to Eq. (2). We have considered other tests, such as $\chi^2$, but these other tests are often based on assumptions that are probably not met in our case. It is for example well known that ln T does not necessarily obey Gaussian statistics in the SAL regime[30].

Next, we attempt to describe the data with SC theory over the full range of frequencies $\nu'$ using a single adjustable parameter ($kl$), as a measure of the distance to the critical point at ($kl$) = 1. We find excellent agreement between SC theory and the data over the entire range $\nu' \neq [0.45, 0.5]$, i.e., outside a central frequency interval in or near the full bandgap, Figs. 5 and 6. The observation of such a single-parameter scaling is the key finding of our work. In the regime where ($kl$) $\lesssim$ 1.2 SC theory describes the data substantially better than diffusion theory and we can describe the data across the critical transition from light diffusion to localization. In the low-frequency branch, between the mobility edge $\nu'_{c,l} = 0.412$ and the band edge $\nu' \simeq 0.47$, the sample

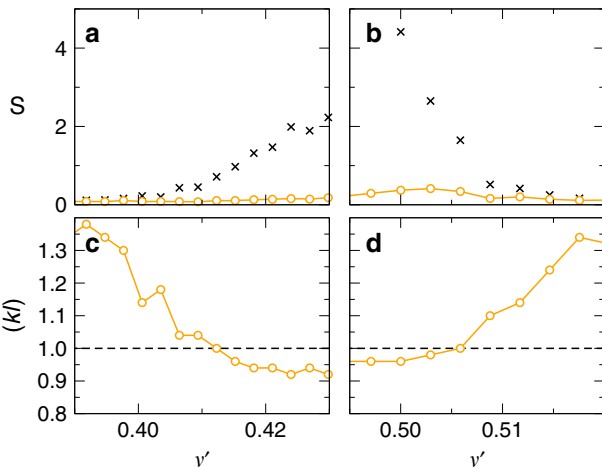

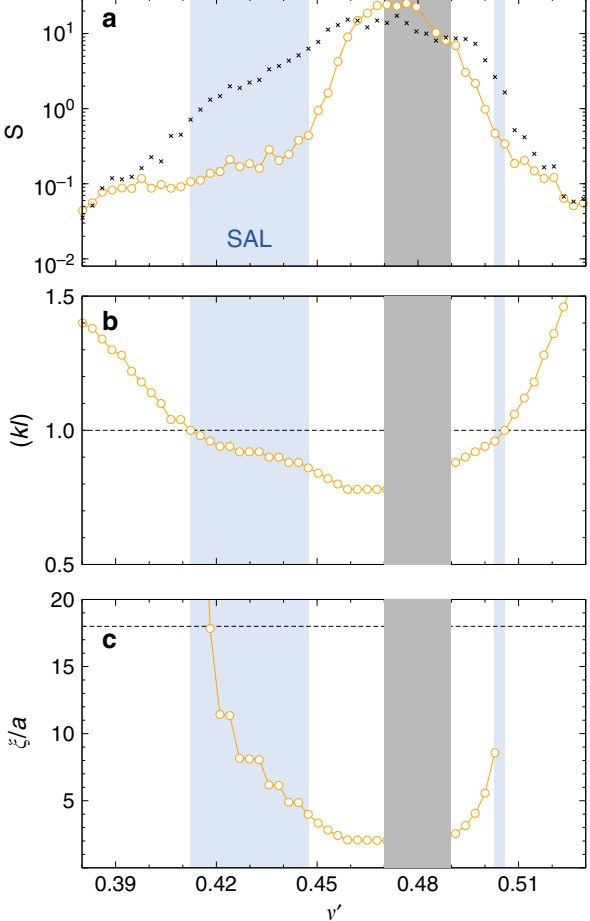

**Fig. 4 Two-parameter fit to the self-consistent (SC) theory of localization.** Fitting error $S$ for a comparison to diffusion theory (crosses) and SC theory (open circles) for frequencies below the gap frequency $\nu_{Gap}$ (**a**) and above (**b**). Adjustable parameters are $(kl)$ and $\ell$. **c**, **d** Localization parameter $(kl)$ extracted from the fit for the same frequency intervals. We find $(kl) = 1$ for $\nu'_{c,l} = 0.412$ ($\ell_{c,l}/a = 0.513$), (**c**), and $\nu'_{c,h} = 0.506$ ($\ell_{c,h}/a = 0.242$), (**d**).

remains localized over a large frequency interval, comparable or larger in width than the bandgap, as shown in Fig. 5. In this regime, the localization parameter drops from $(kl) = 1$ to about $(kl) = 0.85$. In the high-frequency branch, the localized regime is nearly degenerate and the sample rapidly enters the gap after crossing the mobility edge from above for $\nu' < \nu'_{c,h}$. For $\nu' < 0.39$ and $\nu' > 0.52$ the position dependence of $D(z)$ is weak, for our system sizes, and the predictions by SC theory and by diffusion theory are indistinguishable.

## Discussion

In summary, we could show that hyperuniform 3D silicon networks display different characteristic transport regimes for electromagnetic vector waves. Deep in the gap region, the transmittance decays exponentially, indicating tunneling through the entire sample. Outside the gap region, we observe a critical transition from classical diffusion to wave localization controlled by a single parameter $(kl)$. We have shown that in this regime, our numerical data can be described quantitatively by assuming a position-dependent diffusion coefficient $D(z)$ derived from the SC theory of localization. Finding such an agreement is generally understood as evidence for SAL of light. Moreover, our detailed results about the transition to SAL in realistic digital representations of dielectric networks can provide valuable guidance for future experimental attempts to probe light localization.

Finally, we would like to add a remark concerning the breakdown of the description of wave transport by SC theory and the opening of the gap. SC theory is a heuristic approach that postulates a position-dependent diffusion coefficient $D(z)$ but does not provide a microscopic explanation (which, under certain conditions has been added later[38]). The concept of applying a diffusion equation to describe transport over certain distances $\xi > \ell$ breaks down as $\xi/\ell \to 1$. It it thus unclear whether evanescent decay, Fig. 1d, and the correspondingly poor fit with SC theory, within $\Delta\nu' \simeq 0.01 - 0.02$ next to the band edge, is due to the breakdown of SC theory or other emerging transport phenomena in the vicinity of the gap, as suggested earlier in ref. [30]. Moreover, we find it interesting to speculate whether the SC theory could be generalized to provide a unified theory for wave transport in

**Fig. 5 Single-parameter fit to the data using SC theory.** The values obtained from the fit are displayed as open circles. The only adjustable parameter is $(kl)$. The optical mean free path $\ell$ is defined via $(kl) = \ell/\ell_c \times \nu'/\nu'_c$, Fig. 4. **a** Least square error $S$. Crosses: Fit to the data with diffusion theory, Eq. (1) with $\tilde{L} = L$. The blue shaded area indicate where the SC theory provides an accurate and better description of the numerical data indicating SAL. **b** Localization parameter extracted from the single-parameter fit. **c** The localization length, given by $\xi/a = 6(\ell/a)(kl)^2/(1 - (kl)^4)$. The horizontal dashed line indicates the largest accessible length scale in the simulation, given by the edge length of the simulation box $L = 18a$.

amorphous photonic materials encompassing the transition between diffusive, localized, and bandgap regimes.

## Methods

**FDTD simulations.** FDTD simulations were performed using the MEEP software[49] and were run on a computer cluster. Throughout this work, the Poynting vector is recorded on a monitor situated behind the structure. Transmittance is defined as the ratio of the transmitted power to the incoming one and is calculated by dividing the transmitted power (integral of the Poynting vector over the monitor) by the power transmitted in a reference run (empty simulation box). The network structures were generated using a custom-made code (MATLAB and Statistics Toolbox, The MathWorks, Inc., Massachusetts, United States) based on the full 10,000 particle seed pattern taken from[48]. Equivalently, the sicipy.spatial open-source library of Python can be used for this purpose. Using a clean cut we obtain slabs of different thickness $L \le 18a$, which were then imported to MEEP (see Table 1 for the exact values of thickness). All units were set to μm and the sphere diameter was set to $a = 5/3$ μm. We apply periodic boundary conditions in x-and y-directions. Since we do not know the precise size of the simulation box used in ref. [48], the periodic boundary conditions in MEEP will not exactly match the original periodic boundary conditions employed when generating the pattern. From our own band structure calculations we find that this can give rise to a few

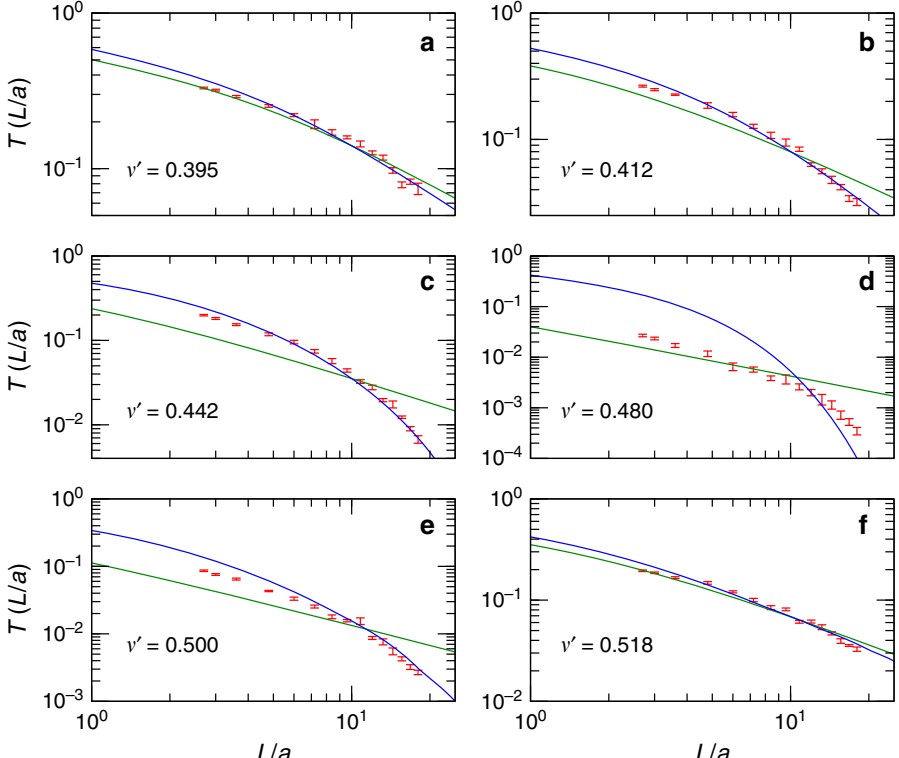

**Fig. 6 Transmittance $T$ as a function of the slab thickness.** Transmittance $T$ as a function of $L/a$ in log–log representations for selected frequencies (**a**–**f**). Silicon volume fraction $\phi = 0.28$. Only the averaged data for ($L/a \geq 2.7$), shown in these plots, were taken into account for fitting. Error bars denote the standard deviation of the results obtained from 15 (thin samples) to 6 (thick samples) realizations. Blue lines: best fit with SC theory and Eq. (1). Green lines: best fit with Eq. (1) for ($kl$) → ∞, in the absence of SAL.

defect states in the gap. We do not expect these defect states to contribute to transport in the diffuse or SAL regime, where the DOS is high, but they can increase the transmittance in the gap for $L \gg L_B$ due to tunneling between defect states. We believe this is the main reason why $T(L) > \exp(-L/L_B)$ for $L \gg L_B$ in the gap regime, see e.g., Fig. 6d. Since the gap regime is not in the focus of our study, we have not explored this in more detail but plan to address this in future work, using new seed pattern properly matched to the periodic boundary conditions of the MEEP-simulation box.

The network was illuminated with a broadband pulse of linearly polarized light with electric field vector parallel to one of the sides of the simulation box. The pulse bandwidth was sufficient to cover vacuum wavelengths between 1 and 7 μm corresponding to reduced frequencies $\nu' \in [0.24, 1.67]$. Perfectly matched layers (PML) were fitted at both ends of the simulation box and periodic boundary conditions were applied perpendicular to the wave propagation direction. The PML's absorb all transmitted and reflected waves (regardless of incidence direction) and prevent them from re-entering the simulation box. The PML thickness was 7 microns, which ensured that all wavelengths shorter than this value were suppressed. The spatial resolution was equal to 20 pixels per μm. Convergence tests were performed to check the robustness of the simulation. It was verified that increasing the spatial resolution by a factor of two did not considerably influence the transmittance curves. Also the simulation time was selected in a way to yield robust results. By placing an additional monitor between the source and the network, we checked for flux conservation over the entire frequency interval of interest.

**Numerical implementation of the SC theory of localization.** In the self-consistent theory of localization, the standard diffusion equation is replaced by an equation where the diffusion coefficient is nonlocal both in the space and time domain. The renormalization of the diffusion coefficient accounts for the different return probability when interference effects are considered in the multiple scattering regime[38]. We are considering a slab geometry and continuous wave illumination at a given carrier wave frequency. The simplified geometry, invariant in the plane parallel to the slab, leads to a set of SC equations for the diffusion constant dependence and the diffusion equation Green's function $g(z, z')$ in the direction $z$ perpendicular to the slab surfaces (lying between $z = 0$ and $z = L$). Here we work in reduced units, where all lengths are scaled by the mean free path $\ell$. The diffusion coefficient $D(z)$ is normalized by the Boltzmann diffusion coefficient $D_B$. Taking the Fourier transform in the $x$, $y$ plane, parallel to the slab boundaries, we

reach at the diffusion equation:

$$-\frac{\partial}{\partial z}\left[D(z)\frac{\partial}{\partial z}g(q,z,z')\right] + q^2 D(z)g(q,z,z') = \delta(z - z'), \quad (3a)$$

$$g(q,z=0,z') - z_0 D(z=0)\frac{\partial}{\partial z}g(q,z,z')\bigg|_{z=0} = 0, \quad (3b)$$

$$g(q,z=L,z') + z_0 D(z=L)\frac{\partial}{\partial z}g(q,z,z')\bigg|_{z=L} = 0. \quad (3c)$$

This result together with the self-consistent equation for the diffusion coefficient

$$D(z) = \left[1 + \frac{3}{(kl)^2}\int_0^{q_{max}^2} d(q^2)g(q,z,z)\right]^{-1}, \quad (4)$$

determines the transport properties of our system for all values of the parameters. In the above equation the cut off is given by $q_{max} = 1/3(kl)_c^2$. The latter depends on the chosen value $(kl)_c$ with $(kl) = (kl)_c$ at the mobility edge.

We solve the set of self-consistent equations by recursively solving Eq. (3) in a first step for all positions of the source $z'$ and transversal wavenumbers $q$. The solution of the Green function is then plugged into Eq. (4) to correct for the $z$-dependent diffusion coefficient $D(z)$ which is inserted back to Eq. (3) until convergence is reached. We choose a discretization scheme where all the considered positions $z$ and wavenumbers squared $q^2$ are evenly spaced. Depending on the values of the parameters $(kl)$, $(kl)_c$, and total length $L$, we need to take a step $\Delta z \equiv h$ ranging from 0.02 to 0.2 and between 300 and 600 steps in $q^2$ to achieve a relative precision in $D(z)$ of the order or $10^{-4}$ in 5–50 recursion steps.

Taking the second order finite differences approximation for the derivatives in Eq. (3) leads to a tridiagonal system of equations which has to be solved for each value of the wavenumber $q$ and position of the source. Obviously, changing the position of the source amounts to changing the independent term of the system of equations and hence all equations for a given value of $q$ can be solved at once through the inverse of the corresponding tridiagonal matrix. We choose the Lapack function DGTSV to get the inverse since it is simple to use and universally accessible while efficient enough for our purposes[53]. Specifically, if we take a discretization $z_i = h(i - 1)$, for $i = 1, \cdots, n$. And naming $D_i \equiv D(z = z_i)$,

$D'_i \equiv \frac{dD(z=z_i)}{dz}$, the diagonal terms of the system of equations read

$$\text{diag}_1 = 1 + \frac{h}{z_0 D_1}, \tag{5a}$$

$$\text{diag}_{i=2,\cdots,n-1} = D_i \left(2 + h^2 q^2\right), \tag{5b}$$

$$\text{diag}_n = 1 + \frac{h}{z_0 D_n}, \tag{5c}$$

the subdiagonal terms are

$$\text{subdiag}_{i=1,\cdots,n-2} = \frac{D'_{i+1} h}{2} - D_{i+1}, \tag{6a}$$

$$\text{subdiag}_{n-1} = -1. \tag{6b}$$

Analogously, the superdiagonal terms are

$$\text{superdiag}_1 = -1, \tag{7a}$$

$$\text{superdiag}_{i=2,\cdots,n-1} = -\frac{D'_i h}{2} - D_i. \tag{7b}$$

Since the sources are located in the interior of the slab, the points at which the source is located are $z'_{i=2,\cdots,n-1} = h(i-1)$. The independent term vector $T_i$ for the source at $z_i$ is $(T_i)_j = h\delta_{i,j}$.

Once the numerical solution is obtained for all the source positions, the value of $g(z_i, z_i)$ is available for $i = 2, \cdots, n-1$. The $g(z_1, z_1)$ and $g(z_n, z_n)$ are obtained by second order accuracy extrapolation of the solutions $g(z_i, z_i)$ to the boundary. The full solution is then used to calculate the integral in Eq. (4) by 3/8 Simpson's rule. The updated value of the diffusion coefficient $D(z)$ is used in the next step of recursion. The algorithm stops when the logarithm of an update of $D(z)$ differs by $<10^{-4}$ from the previous value at all the points in the discretization. The seed $D(z)$ used to start the recursions is set to 1. In all cases $(kl)_c = 1$.

**Transmission through a slab.** In the following, we describe how to consistently merge SC theory with the theory by Lemieux et al.[52]. The contribution that is affected by SC theory is the total diffusive transmittance $T_d$ (see also[54]). We compute it considering that the sources of diffuse intensity are continuously distributed across the slab with an intensity proportional to the ballistic intensity $I_b = \exp(-z)$. To simplify the notation, in this section, we work in reduced units where all lengths are normalized to $\ell$ and $D(z)$ is normalized to $D_B$. The diffusive transmittance, neglecting boundary reflectivity of a few percent, is hence

$$T_d = \int_0^L T_{SCT}(z)e^{-z}dz, \tag{8}$$

where $T_{SCT}(z)$ is the diffuse transmittance for a source at $z = z'$ (see Supplementary Material, Eqs. (S5)–(S17), for details), explicitly we have

$$T_d = \int_0^L \frac{z_0 + \int_0^z \frac{1}{D(x)}dx}{2z_0 + \bar{L}} e^{-z}dz, \tag{9}$$

with

$$\bar{L} \equiv \int_0^L \frac{1}{D(x)}dx. \tag{10}$$

The first term ($z_0$ in the numerator) can be explicitly integrated:

$$\int_0^L \frac{z_0}{2z_0 + \bar{L}}e^{-z}dz = \frac{z_0}{2z_0 + \bar{L}}\left(1 - e^{-L}\right). \tag{11}$$

The second term in Eq. (9) is more involved since $D(z)$ is not known a priori, the integral to be solved is

$$I \equiv \frac{1}{2z_0 + \bar{L}}\int_0^L e^{-z}\int_0^z \frac{1}{D(x)}dxdz, \tag{12}$$

that can be formally integrated by parts using $u(z) \equiv \int_0^z \frac{1}{D(x)}dx$ and $dv(z) \equiv e^{-z}dz$. Hence

$$I = \frac{1}{2z_0 + \bar{L}}\left[-e^{-L/\ell}\bar{L} + \int_0^L dz \frac{e^{-z}}{D(z)}\right]. \tag{13}$$

The second term in the rhs of Eq. (13) can again be formally integrated by parts using $u(z) \equiv 1/D(z)$, $dv(z) \equiv e^{-z}dz$ to give

$$\int_0^L dz \frac{e^{-z}}{D(z)} = \frac{1}{D(0)}\left(1 - e^{-L}\right) + \int_0^L \frac{e^{-z}}{D(z)}\frac{d\ln[D(z)]}{dz}dz. \tag{14}$$

Collecting results, we have

$$T_d = \frac{z_0 + 1/D(0)}{2z_0 + \bar{L}}\left(1 - e^{-L}\right) - \frac{\bar{L}}{2z_0 + \bar{L}}e^{-L} + \frac{\eta(L)}{2z_0 + \bar{L}}, \tag{15}$$

where

$$\eta(L) \equiv \int_0^L \frac{LN(z)}{D(z)}e^{-z}dz, \text{ and } LN(z) \equiv \frac{d[\ln(D(z))]}{dz}. \tag{16}$$

Finally, adding the ballistic transmittance $e^{-L}$, we obtain Eq. (1) for the total transmittance:

$$T(L) = \frac{z_0 + 1/D(0)}{2z_0 + \bar{L}}\left(1 - e^{-L}\right) - \frac{\bar{L}}{2z_0 + \bar{L}}e^{-L} + \frac{\eta(L)}{2z_0 + \bar{L}} + e^{-L}. \tag{17}$$

It is worth noticing at this point that in the limit of standard diffusion theory, $D(z) = 1$ (i.e. the diffusion coefficients coincides with the standard $D_B = c\ell/3$) and $\eta(L) = 0$, since $LN(z) = 0$.

## Data availability
The data of $T(L/a)$ shown in the paper, either obtained from FDTD calculations or from the self-consistent theory of localization (SC theory) are provided in the online repository (https://doi.org/10.5281/zenodo.3968424). These and all other data sets used in the study can be generated from the codes uploaded to the repository, or can be obtained from us upon reasonable request.

## Code availability
The codes used to produce the results of this study are included in the repository https://doi.org/10.5281/zenodo.3968424 and described in the main text or the supplementary material. With respect to third party codes, such as the open-source codes MPB and MEEP, we refer to the original publications, see refs. [49–51]. Links to the original third party sources are provided in the README.txt files in the repository.

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

## Acknowledgements

We thank Seng Fatt Liew and Hui Cao for discussions and for sharing the DOS data published in ref. [26]. We thank Sergey Skipetrov for discussion and for providing us with the original FORTRAN code to numerically solve the SC-theory equations, which we have implemented and adapted as described in the text and Supplementary material. F.S. thanks Dave Pine for discussion and comments. This research was supported by the Swiss National Science Foundation through the National Centre of Competence in Research *Bio-Inspired Materials* and through project number, 169074, 188494 (F.S.) and 192340 (L.S.F.P.). J.H. acknowledges the use of the TeraACMIN computer cluster at the Academic Centre for Materials and Nanotechnology, AGH-UST, Krakow, Poland.

## Author contributions

F.S. and J.H. designed the study. J.H. carried out the FDTD calculations using MEEP. L.S. F.P. did the band structure calculations using MPB. L.S.F.P. did all the fitting and comparison to SC theory. F.S. wrote the paper with contributions by all authors.

## Competing interests

The authors declare no competing interests.
