## [Peer Review File · Nature Communications]

Reviewers' Comments:

Reviewer #1:

Remarks to the Author:

This is an interesting paper which describes a numerical study of light localization in a random 3D hyperuniform network. This is a timely topic that should be of interest to people in several different communities. The beginning of the paper is beautifully written and nicely describes the historical background and context of light localization, band gaps, and hyperuniformity. Things are quite clear until we get to the bottom of page 8, where the paper descends into incomprehensibility. A position dependent diffusion coefficient is introduced, which has different behavior depending on its vicinity to the mobility edge. Integrating over the thickness of the sample gives L/l (or L -tilde). I find the next several sentences and the bottom of page 8 and top of page 10 utterly opaque. In particular, the sentences, "We introduce (kl) -tilde ...To identify... and fit the prediction of the SC-theory to the data ..." Which data are they talking about? the transmission data in Figure 3(C), or some other data. I also do not understand the significance of the "state point". [Aside: Whatever they are fitting, they report the reduced chi-squared, with values as low as 0.001. Using the normal definition of chi-squared, one should never have values much smaller than unity (see Numerical Recipes, for example). So they need to define better what they are doing.]

I do not understand how the theory is related to the actual measurements they make, which I suppose are the plots of Figure 3C. How can they discuss the chi-squared of the fits (in Fig. (2)) before they even show the fits (in Fig. (3)).

Some help is provided by reading the Supplementary Information, particularly the last paragraph of page 7 where they explain how the value of L -tilde obtained from integrating $D(z)$ (which is obtained by numerical solutions of a system of equations) is substituted into Eq. (1) to obtain the transmission, which can be compared to their data. However, the description in the main text is completely incomprehensible. They need to make the text in the main text stand better on its own. If they cannot, then maybe this work should be published elsewhere.

Reviewer #2:

Remarks to the Author:

The manuscript 'Transition from Light Diffusion to Localization in Three-Dimensional Hyperuniform Dielectric Networks near the Band Edge' by Jakub Haberkov, Luis S. Froufe-Pérez, and Frank Scheffold studies the strong localization of light in an amorphous photonic material near the band gap.

Addressing an intricate problem, the authors use an appealing combination of state of the art software and numerical calculations with recent theoretical insights to study intriguing questions that are important both for applications and from a more fundamental point of view.

The applied concepts, in particular, the amorphous photonic network, are of high relevance and recent interest.

Similar findings of strong Anderson localization has, as the authors point out, been highly debated.

While both the numerical data and the theoretical models are convincing to me, there are important questions that the authors must and hopefully can answer concerning the statistical analysis.

The latter is central to the key finding of the paper, as the authors correctly point out.

#Main questions

* Perhaps it is just a different use of technical terms, but my most important question concerns the 'reduced chi-squared'.

Typically, it should be close to unity for a preferred model that fits the data without overfitting the parameters.

Why are here values preferred that are as small as possible?

* Related to this question, in the Supplementary Material, χ^2 is defined using a logarithmic loss function and without taking a variance of the points into account.

This is an unusual definition, at least to me, since I know the standard chi-squared value as a sum of residuals weighted by the variance, which is justified by a maximum likelihood fit for Gaussian errors.

Can the authors please explain and justify this choice?

Is this loss function used for all fits or only for the SC results?

What is the distribution of the residuals?

Might a Kolmogorov-Smirnov test be useful?

* The degrees of freedom are mentioned to be 15, which I assume corresponds to 15 different values of L .

Are there not 6 to 15 samples per value of L ?

How is the number of free parameters in the non-linear fit estimated?

* For the upper band gap, the authors point out the very short regime of localization, and the chi-squared values of the two fits are almost equivalent. How well established is it that there is an upper localization regime?

Is there an intuitive reason for the asymmetry in the sizes of the regimes of localization?

* How important is the evaluation of the local minima in the chi-squared value to determine $\nu_{c,l}$ and $\nu_{c,h}$?

Due to the statistical fluctuations, it is difficult to judge whether the local minima are actual features. More apparent is the change in the derivative. Is this to what the estimate refers or is not rather the difference to the chi-squared value of the fit to Eq. (1) more important? Will it help to establish the reliability of the values, if the results for different values of the critical value are included in the Supplementary Material?

* It has been difficult for me to understand the exact fitting procedures (when and why parameters were chosen to be either fixed or free) and thus evaluate the analysis.

Can the authors please extend and clarify the presentation?

For example, it would be helpful for me, if the dependencies of the transmission were clearly expressed by writing it as a function of all of its parameters.

#Minor remarks

* How is the central frequency of the gap defined? How is its width defined?

From the text, I assume that the edges are determined independently from the diffusion analysis.

* For reproducibility, the authors should provide more details about the samples, e.g., the number of spheres per sample, the simulation procedure and its parameters, the boundary conditions in z -direction.

Will the samples and statistical analysis be published alongside the values of the transmission?

* The threshold value to identify the classical diffusive regime is stated as 5×10^{-3} . From Figure 2E and S3B, it appears to be 3×10^{-3} .

* Besides the statistical analysis, the overall presentation is clear (both in text and figures). My only suggestion is to define symbols immediately when they are used the first time (e.g., z_0 and D_B).

* Just for the authors' interest, two typos: Figure 2 'the the best fit', Figure 3 'Eq.((1)'

Hoping that my questions are helpful, I look forward to the authors' reply.

Reviewer #3:

Remarks to the Author:

The authors analyse theoretically the possibility of obtaining Anderson localization in a disordered system with a particular type of design. They show that Anderson localisation is possible in these types of materials, which is an enormous result given the current state-of-the-art of the field. After theoretical and experimental analysis in recent years it became doubtful if the phenomenon could exist at all (for light) in three dimensions.

The presented results look sound and the group has an excellent reputation in the field. I think the results can be trusted, which means that there is a large breakthrough now.

A next step would be a convincing experimental verification in a material like that presented in this paper.

There is one minor remark from my side, which is on terminology. The authors keep using the word hyper-uniform, while they themselves have shown that hyperuniformity as such is not an important aspect. In particular, as far as I understood, a hyper uniform structure does not seem to have any advantage in terms of bandgap with respect to a structure with traditional correlations. Hyperuniformity seems just one of many ways to get the short range correlations needed for the bandgap and hence localization. Using the terminology this way risks of increasing the (already present) hype regarding hyper uniformity.

Having said that, apart from this question on terminology, I find the results beautiful and think they will have a broad impact. Also the paper is exceptionally well written and clear. Can recommend publication in Nature Communications.

Reviewer #4:

Remarks to the Author:

I hereby provide comments on the manuscript "Transition from Light Diffusion to Localization in Three-Dimensional Hyperuniform Dielectric Networks near the Band Edge" by Haberko et al, submitted for publication in Nature Communications.

The paper presents a numerical study of light transport in a correlated disorder structure ("hyperuniform network) and shows that the transition to localization at the mobility edge proceeds in agreement with the self-consistent theory of localization (diffusion coefficient dependent on position).

The localization of electromagnetic waves is a very interesting topic and its experimental

observation has been subject to certain controversy as the authors note. The main findings of the manuscript include the spectral identification of the mobility edges and the fact that a single parameter, $|\tilde{k}|$, can describe the transmission across the critical region where the transport transitions diffusion to localization.

Some minor points:

- presentation can be improved especially the motivation about the present study (most of it dwells on experimental observations of light localization and absence of light localization in point scatterer ensemble, none of which are subject of the current study)
- some typos and cumbersome formulations need to be corrected (page 2: "The same time"; page 4: "derived by a Delaunay", page 12: "we could show, that"; etc.)

Some major points:

In general, I find difficult to identify a significant advance in understanding the physics of the localization phenomena that is likely to influence the field. The self-consistent theory of localization has been validated in 3D elastic networks [1], numerical studies of light localization in disordered photonic structures have been performed [2], [3] (works surprisingly not cited in the current manuscript).

There are some issues that the authors need to have addressed:

- The authors employ a "hyperuniform" photonic network. It is unclear what is the role of the "hyperuniformity" in their study. Moreover, the disorder in their network is not quantified (how hyperuniform the network? how far/close is to an ordered/random network?). Is there a notable difference between the network used and a perturbed periodic network (as the one used in [2])? Are their conclusions valid for other disordered networks? What is the influence of the correlations in the structure on the localization?
- The point raised above is highly relevant in the context of the methodology used by the authors. Most of their conclusions are based on statistical analysis of the transmission data: fitting to obtain the z_0 variable, χ^2 analysis to identify the mobility edges etc. I find the approach rather phenomenological and there are places where some serious doubts about the generality of the conclusions drawn can be raised
 - o non-zero DOS in the gap accompanied by transmission on the order of 1% (is there a true full band gap?)
 - o imprecise determination (about plus minus 15%) of z_0 (3.26 ± 0.5) and R (0.66 ± 0.05);
 - o rather small system size (18 x average scatterer distance) which makes the transmission analysis more complex and requires approximations further compounding the validity of the results; a larger sample would have allowed a direct verification of the self-consistent theory prediction in identifying the mobility edge as $\nu = \nu_c$
 - o rather small number of ensemble members (15 thin slabs and only 6 thick slabs)
- The absence of the analysis of the spatial extend of the band-tail states that could have clearly singled the localization transition at the mobility edge. I would strongly recommend a methodology similar to the one used in [4], where convincing evidence (both theoretical/computational and experimental) is presented for localization of light in 2D compositionally disordered structures.
- The authors use a computational mesh with 20 points/micron (about 33 pixels per a). They claim that increasing the resolution "did not considerably influence" the transmission results. I don't find this convincing, as they would need to check that the results for the mobility edge predictions and the delicate statistical analysis from Figures 2 and 3 are not influenced by the resolution used.

[1] H Hu, A Strybulevych, JH Page, SE Skipetrov, BA van Tiggelen, "Localization of ultrasound in a three-dimensional elastic network", Nature Physics 4 (12), 945

- [2] C. Conti, A. Fratalocchi, Dynamic light diffusion three-dimensional Anderson localization and lasing in inverted opals, *Nat. Phys.*, Vol. 4, 2008, pp. 794-798. [28].
- [3] D. Molinari and A. Fratalocchi, "Route to strong localization of light: the role of disorder," *Opt. Express* 20, 18156-18164 (2012).
- [4] M. Lee, J. Lee, S. Kim, S. Callard, C. Seassal and H. Jeon, "Anderson localizations and photonic band-tail states observed in compositionally disordered platform" , *Science Advances* 4, e1602796 (2018).

Summary of changes made:

- Page 4/5: Added more details about SC-theory and its range of validity and application to our system
- Page 7: Added a section explaining in more detail our fitting procedure. We replaced $\tilde{\kappa}$ by κ and ℓ (LaTeX notation) as explained in the text.
- To find the mobility edge we now perform a two parameter fit with κ and ℓ which, as we show in the new Figure 3, is more convincing and robust compared to the original approach (see original Figure 2E).
- We have significantly expanded the methods section.
- Additionally, we have made a few minor changes to correct typos or improve the text for clarity.

Figures

We have rearranged the panels of the original Figure 1 and 2. Some of the original plots have been removed owing to the new fitting procedure.

- New Figure 1: Added results of our own band gap calculation (using MPB) in panel (b) as explained on page 7. A comparison to the results by Liew et al., shown previously, is given in Figure S1.
- New Figure 1: We modified panel (d) using colored symbols for clarity.
- New Figure 2: Same content as Figure 2B and part of 2E in the original manuscript.
- New Figures 4 and 5 (single parameter fit and plot $T(L)$) replace the old Figure 3

Next page: Reply to the Reviewers' comments- NCOMMS-

Reviewers' comments:

Reviewer #1 (Remarks to the Author):

This is an interesting paper which describes a numerical study of light localization in a random 3D hyperuniform network. This is a timely topic that should be of interest to people in several different communities. The beginning of the paper is beautifully written and nicely describes the historical background and context of light localization, band gaps, and hyperuniformity.

We would like to thank the reviewer for the general positive assessment up to this point and will address the more critical comments below.

Things are quite clear until we get to the bottom of page 8, where the paper descends into incomprehensibility. A position dependent diffusion coefficient is introduced, which has different behavior depending on its vicinity to the mobility edge. Integrating over the thickness of the sample gives L/l (or L -tilde). I find the next several sentences and the bottom of page 8 and top of page 10 utterly opaque. In particular, the sentences, "We introduce (kl) -tilde ...To identify... and fit the prediction of the SC-theory to the data ..." Which data are they talking about? the transmission data in Figure 3(C), or some other data. I also do not understand the significance of the "state point". [Aside: Whatever they are fitting, they report the reduced chi-squared, with values as low as 0.001. Using the normal definition of chi-squared, one should never have values much smaller than unity (see Numerical Recipes, for example). So they need to define better what they are doing.]

I do not understand how the theory is related to the actual measurements they make, which I suppose are the plots of Figure 3C. How can they discuss the chi-squared of the fits (in Fig. (2)) before they even show the fits (in Fig. (3)).

Some help is provided by reading the Supplementary Information, particularly the last paragraph of page 7 where they explain how the value of L -tilde obtained from integrating $D(z)$ (which is obtained by numerical solutions of a system of equations) is substituted into Eq. (1) to obtain the transmission, which can be compared to their data. However, the description in the main text is completely incomprehensible. They need to make the text in the main text stand better on its own. If they cannot, then maybe this work should be published elsewhere.

We acknowledge that the fitting procedure is somewhat complicated. In the revised manuscript we have made a number of changes to the fitting procedure and we also describe the procedure more clearly.

Reviewer #2 (Remarks to the Author):

The manuscript 'Transition from Light Diffusion to Localization in Three-Dimensional Hyperuniform Dielectric Networks near the Band Edge' by Jakub Haberko, Luis S. Froufe-Pérez, and Frank Scheffold studies the strong localization of light in an amorphous photonic material near the band gap. Addressing an intricate problem, the authors use an appealing combination of state of the art software and numerical calculations with recent theoretical insights to study intriguing questions that are important both for applications and from a more fundamental point of view. The applied concepts, in particular, the amorphous photonic network, are of high relevance and recent interest.

Similar findings of strong Anderson localization has, as the authors point out, been highly debated. While both the numerical data and the theoretical models are convincing to me, there are important questions that the authors must and hopefully can answer concerning the statistical analysis. The latter is central to the key finding of the paper, as the authors correctly point out.

#Main questions

* Perhaps it is just a different use of technical terms, but my most important question concerns the 'reduced chi-squared'. Typically, it should be close to unity for a preferred model that fits the data without overfitting the parameters. Why are here values preferred that are as small as possible?

* Related to this question, in the Supplementary Material, χ^2 is defined using a logarithmic loss function and without taking a variance of the points into account. This is an unusual definition, at least to me, since I know the standard chi-squared value as a sum of residuals weighted by the variance, which is justified by a maximum likelihood fit for Gaussian errors. Can the authors please explain and justify this choice? Is this loss function used for all fits or only for the SC results? What is the distribution of the residuals? Might a Kolmogorov-Smirnov test be useful?

* The degrees of freedom are mentioned to be 15, which I assume corresponds to 15 different values of L . Are there not 6 to 15 samples per value of L ? How is the number of free parameters in the non-linear fit estimated?

We would like to thank the referee for pointing this out. In the original manuscript, our notation has not been very consistent. In the revised version we clearly refer to our procedure as a least-square fitting of $\ln T$, the most simple and straightforward approach.

We have studied other tests but, as pointed out by the referee, these other tests are based on assumptions that are probably not met in our case. It is for example well known that $\ln T$ does not obey Gaussian statistics in the SAL regime (see e.g. Froufe-Pérez et al. PNAS 114, 9570 (2017)).

* For the upper band gap, the authors point out the very short regime of localization, and the chi-squared values of the two fits are almost equivalent. How well established is it that there is an upper localization regime? Is there an intuitive reason for the asymmetry in the sizes of the regimes of localization?

This is an interesting question that we currently cannot answer. We have applied the same procedure to the upper and lower frequency regime and did not want to conceal this data. Our analysis shows that, in the upper localization regime, SC theory can describe the data better than diffusion theory over significant range of frequencies (already for $kl > 1$, approaching the mobility edge).

* How important is the evaluation of the local minima in the chi-squared value to determine $\nu'_{c,l}$ and $\nu'_{c,h}$? Due to the statistical fluctuations, it is difficult to judge whether the local minima are actual features. More apparent is the change in the derivative. Is this to what the estimate refers or is not rather the difference to the chi-squared value of the fit to Eq. (1) more important? Will it help to establish the reliability of the values, if the results for different values of the critical value are included in the Supplementary Material?

We agree that this fitting procedure was not very robust. We have replaced this fit with a two parameter fit to determine $kl=1$ (see text) which is much more robust and convincing we believe. Nonetheless, the values of ν'_c we find, using the new procedure, are almost identical to the one reported in the original manuscript.

* It has been difficult for me to understand the exact fitting procedures (when and why parameters were chosen to be either fixed or free) and thus evaluate the analysis. Can the authors please extend and clarify the presentation? For example, it would be helpful for me, if the dependencies of the transmission were clearly expressed by writing it as a function of all of its parameters.

We have added a section on the fitting procedure and also improved the text for clarity.

#Minor remarks

* How is the central frequency of the gap defined? How is its width defined?

From the text, I assume that the edges are determined independently from the diffusion analysis.

For the revised manuscript we redid the PBG calculation using the MPB package and find a sharp band edge. The definitions are now clear.

* For reproducibility, the authors should provide more details about the samples, e.g., the number of spheres per sample, the simulation

procedure and its parameters, the boundary conditions in z -direction. Will the samples and statistical analysis be published alongside the values of the transmission?

We have added this information and also have added error bars to the points. All original MEEP data will be made available on a repository. We obtain the finite sized slab using periodic boundary conditions in x - and y - direction and a clean cut in z -direction.

* The threshold value to identify the classical diffusive regime is stated as 5×10^{-3} . From Figure 2E and S3B, it appears to be 3×10^{-3} .

* Besides the statistical analysis, the overall presentation is clear (both in text and figures). My only suggestion is to define symbols immediately when they are used the first time (e.g., z_0 and D_B).

* Just for the authors' interest, two typos: Figure 2 'the the best fit', Figure 3 'Eq.((1)'

Hoping that my questions are helpful, I look forward to the authors' reply.

The last points have been addressed or to not apply any more after revisions.

Reviewer #3 (Remarks to the Author):

The authors analyse theoretically the possibility of obtaining Anderson localization in a disordered system with a particular type of design. They show that Anderson localisation is possible in these types of materials, which is an enormous result given the current state-of-the-art of the field. After theoretical and experimental analysis in recent years it became doubtful if the phenomenon could exist at all (for light) in three dimensions.

The presented results look sound and the group has an excellent reputation in the field. I think the results can be trusted, which means that there is a large breakthrough now.

A next step would be a convincing experimental verification in a material like that presented in this paper.

There is one minor remark from my side, which is on terminology. The authors keep using the word hyper-uniform, while they themselves have shown that hyperuniformity as such is not an important aspect. In particular, as far as I understood, a hyper uniform structure does not seem to have any advantage in terms of bandgap with respect to a structure with traditional correlations. Hyperuniformity seems just one of many ways to get the short range correlations needed for the bandgap and hence localization. Using the terminology this way risks of increasing the (already present) hype regarding hyper uniformity.

We acknowledge the point raised by the referee. We have replaced 'hyperuniform' by 'amorphous' in the title. We still use it elsewhere since we believe it's a useful concept for a disorder metric and by now it has gained some recognition in the field of photonic bandgap materials.

Having said that, apart from this question on terminology, I find the results beautiful and think they will have a broad impact. Also the paper is exceptionally well written and clear. Can recommend publication in Nature Communications.

Reviewer #4 (Remarks to the Author):

I hereby provide comments on the manuscript "Transition from Light Diffusion to Localization in Three-Dimensional Hyperuniform Dielectric Networks near the Band Edge" by Haberko et al, submitted for publication in Nature Communications.

The paper presents a numerical study of light transport in a correlated disorder structure ("hyperuniform network) and shows that the transition to localization at the mobility edge proceeds in agreement with the self-consistent theory of localization (diffusion coefficient dependent on position).

The localization of electromagnetic waves is a very interesting topic and its experimental observation has been subject to certain controversy as the authors note. The main findings of the manuscript include the spectral identification of the mobility edges and the fact that a single parameter, \tilde{k} , can describe the transmission across the critical region where the transport transitions diffusion to localization.

Some minor points:

-presentation can be improved especially the motivation about the present study (most of it dwells on experimental observations of light localization and absence of light localization in point scatterer ensemble, none of which are subject of the current study)

We do not agree with the referee. Previous experimental studies have been wrong or inconclusive. Due to the technical and experimental difficulties, numerical studies on realistic representations of real-world materials might be the best way to make real progress in this field.

- some typos and cumbersome formulations need to be corrected (page 2: "The same time"; page 4: "derived by a Delaunay", page 12: "we could show, that"; etc.)

We have revised the text and tried to eliminate errors.

Some major points:

In general, I find difficult to identify a significant advance in understanding the physics of the localization phenomena that is likely to influence the field. The self-consistent theory of localization has been validated in 3D elastic networks [1], numerical studies of light localization in disordered photonic structures have been performed [2], [3] (works surprisingly not cited in the current manuscript).

We do not agree with the referee. Due to the technical and experimental difficulties, numerical studies on realistic representations of real-world materials might be the best way to make real progress in this field. Transport in elastic networks considered scalar waves and not vector waves which is known to be critically important. We have cited [2] and believe it's important. Localization in disordered crystals however is qualitatively different to localization in genuinely amorphous materials.

There are some issues that the authors need to have addressed:

- The authors employ a “hyperuniform” photonic network. It is unclear what is the role of the “hyperuniformity” in their study. Moreover, the disorder in their network is not quantified (how hyperuniform the network? how far/close is to an ordered/random network?). Is there a notable difference between the network used and a perturbed periodic network (as the one used in [2])? Are their conclusions valid for other disordered networks? What is the influence of the correlations in the structure on the localization?

For the protocol to generate the hyperuniform amorphous networks we refer to the literature, ‘as described earlier in ref. [23,46]’. We have added two sentences for clarity:

“The design protocol consists of mapping the seed pattern into tetrahedrons by performing a Delaunay tessellation. Then, the centers of mass of the tetrahedrons are connected, resulting in a tetravalent network structure of interconnected rods with the desired structural properties. “

- The point raised above is highly relevant in the context of the methodology used by the authors. Most of their conclusions are based on statistical analysis of the transmission data: fitting to obtain the z_0 variable, χ^2 analysis to identify the mobility edges etc. I find the approach rather phenomenological and there are places where some serious doubts about the generality of the conclusions drawn can be raised
 - o non-zero DOS in the gap accompanied by transmission on the order of 1% (is there a true full band gap?)

We have replaced this figure - see new Figure 1 (b).

- o imprecise determination (about plus minus 15%) of z_0 (3.26 ± 0.5) and R (0.66 ± 0.05);
- o rather small system size (18 x average scatterer distance) which makes the transmission analysis more complex and requires approximations further compounding the validity of the results; a larger sample would have allowed a direct verification of the self-consistent theory prediction in identifying the mobility edge as $\nu = \nu_c$

We had to work with the always limited computational resources and the statistical accuracy available. The data has been compiled from weeks of runs on a substantial part of the Krakow computer cluster.

- o rather small number of ensemble members (15 thin slabs and only 6 thick slabs)

We were initially limited by the seed pattern taken from ref. [47]. We agree that a larger number would have been desirable but we also think that the statistical accuracy of our data is good. We have added error bars to the data points to illustrate this point.

- The absence of the analysis of the spatial extent of the band-tail states that could have clearly singled the localization transition at the mobility edge. I would strongly recommend a methodology similar to the one used in [4], where convincing evidence (both theoretical/computational and experimental) is presented for localization of light in 2D compositionally disordered structures.

We thank the referee for pointing this out. This type of analysis is currently not implemented. We will consider it in the future but for the current manuscript this would be beyond the scope of our work.

- The authors use a computational mesh with 20 points/micron (about 33 pixels per a). They claim that increasing the resolution “did not considerably influence” the transmission results. I don’t find this convincing, as they would need to check that the results for the mobility edge predictions and the delicate statistical analysis from Figures 2 and 3 are not influenced by the resolution used.

We carefully checked the convergence for some examples. Given the limited computational resources we were not able to run the full data set at different resolutions. We have reported this transparently.

[1] H Hu, A Strybulevych, JH Page, SE Skipetrov, BA van Tiggelen, “Localization of ultrasound in a three-dimensional elastic network”, Nature Physics 4 (12), 945

[2] C. Conti, A. Fratalocchi, Dynamic light diffusion three-dimensional Anderson localization and lasing in inverted opals, Nat. Phys., Vol. 4, 2008, pp. 794-798. [28].

[3] D. Molinari and A. Fratalocchi, "Route to strong localization of light: the role of disorder," Opt. Express 20, 18156-18164 (2012).

[4] M. Lee, J. Lee, S. Kim, S. Callard, C. Seassal and H. Jeon, “Anderson localizations and photonic band-tail states observed in compositionally disordered platform” , Science Advances 4, e1602796 (2018).

Reviewers' Comments:

Reviewer #1:

Remarks to the Author:

This is a vastly improved manuscript from the previous version. I believe the work is exciting, important, and of interest to anyone interested in Anderson localization of light. I recommend publication.

However, the explanation of the data analysis is hard to follow and not merely because the analysis is complex. I believe the text can be significantly improved without a great deal of effort.

First, in writing down Eq. (1), it should be explicitly stated that this is not a result taken directly from Durian et al (reference 28), as the text before Eq. (1) states, but one that has been adapted so that it can be used to describe normal light diffusion outside the of the localization region. The authors may not realize it, but they don't actually tell the reader that. If the reader is very persistent and patient, the reader will eventually learn that Eq. (1) is derived in the Methods section. But the poor reader is not told that when Eq. (1) is introduced but is left to think that it's Durian's equation, which is doubly puzzling as Durian did not consider localization at all.

Once they have amended their text to make it clear that Eq. (1) does not come from Durian, but from the Methods section, they can (and should) point out that this more general equation reduces to Durian's result for normal light diffusion by setting $\tilde{L}=L$, $D(0)=DB$, and $\eta=0$.

The discussion of Eq. (2) immediately following Eq. (2) is also hard to follow because the text states that z_0 and l are the only two fitting parameters, but kl appears in the equation. What happened to kl (it disappears in the diffusion limit in which $D(0)$ is DB)? Please alert the gentle reader.

For frequencies within the localization region, I'm not sure what is done. Is $D(0) = DB/(1-(kl)^2)$? If so, it would be helpful to explicitly say so here. If this is what is being done, then equation $D(0) = DB/(1-(kl)^2)$, introduced on page 5, should be set off and given an equation number so the reader more readily understands what is being done here. This is confusing because on page 5 it is stated that this equation is only valid when $kl \gg 1$, which is violated within the localization region.

Finally, I would like to see a plot of $D(z)$, since it plays such a key role in the analysis.

Reviewer #2:

Remarks to the Author:

The authors have thoughtfully answered the referees' questions, and they have carefully and thoroughly revised the manuscript, including a new statistical analysis.

It is a strong improvement that answers all of my essential questions.

Since I am no expert in the self-consistency theory,

I instead focus in my review on the statistical analysis.

Restricting the analysis to a least-square fit is a valid option, and I understand that a statistical hypothesis test is outside the scope of this paper.

In that respect, the authors' answer in the rebuttal, which mentions the non-Gaussian statistics, is important and worth being included in the main text, for example, in an outlook to future research.

The necessary approximations and the limitations of the complex fitting

procedure make it difficult to assess whether the results are a proof or a strong indication of Anderson localization. In either case, the results deserve publication in Nature Communications, where my suggestion is to use a careful wording.

Minor questions and suggestions:

- * On page 7: "that do not explicitly depend on ν ". Since the mobility edges are reduced frequencies, does this statement refer to the three different regimes that are studied?
- * On page 9: Is there an intuitive interpretation of the "extrapolation length"? There is no definition when it first appears in the text.
- * On page 10: The parentheses around k_l are missing in Eqs. (1) and (2).
- * On page 10: Which parameters in Eq. (1) do explicitly depend on k_l ?
- * On page 10: The arguments of T_{SC} in Eq. (2) do not match the definition in Eq. (1).
- * On page 10: "the number is kept constant at $N = 15$ ". What does this mean?
- * On pages 11 and 13: Please use a synonym of the word "significantly" to avoid confusion with its rigorous meaning in statistics. In what sense is the word used on pages 5 and 10?
- * Page 29: Does the "proprietary software library" refer to MATLAB? Maybe it is worth emphasizing here that the main programs used here (MEEP and MPB) are open source software.
- * What is the estimator of the filling fraction? The volume fraction of the original 3D network, or the estimate of MEEP/MPB?
- * What software was used for the Delaunay tessellation?
- * Supplementary Material, page 5: Typo "for both and planes"
- * Supplementary Material, page 6: Typos "abut", "fr", Figure 2 (3 times)

Reviewer #4:

None

Summary of changes made:

The most significant changes to the text are shown on pages 10-13 in response to the remarks by referee #1. We have considered these remarks and explained the origins of Eq. (1) more clearly and also adapted the notation slightly. We would to thank referee #1 for pointing this out and we fully agree with his/her suggestions.

Most other changes are minor language and typographical corrections.

All relevant changes are highlighted in blue color.

Figures:

We have added Figure 3, showing examples of $D(z)$ as requested by referee #1.

Tables:

We have included the exact values of L/a studied numerically as a table in the methods section.

Next page: Reply to the Reviewers' comments

Reviewer #1 (Remarks to the Author):

This is a vastly improved manuscript from the previous version. I believe the work is exciting, important, and of interest to anyone interested in Anderson localization of light. I recommend publication.

We would like to thank the reviewer for the positive assessment.

However, the explanation of the data analysis is hard to follow and not merely because the analysis is complex. I believe the text can be significantly improved without a great deal of effort.

First, in writing down Eq. (1), it should be explicitly stated that this is not a result taken directly from Durian et al (reference 28), as the text before Eq. (1) states, but one that has been adapted so that it can be used to describe normal light diffusion outside the of the localization region. The authors may not realize it, but they don't actually tell the reader that. If the reader is very persistent and patient, the reader will eventually learn that Eq. (1) is derived in the Methods section. But the poor reader is not told that when Eq. (1) is introduced but is left to think that it's Durian's equation, which is doubly puzzling as Durian did not consider localization at all.

Once they have amended their text to make it clear that Eq. (1) does not come from Durian, but from the Methods section, they can (and should) point out that this more general equation reduces to Durian's result for normal light diffusion by setting $\tilde{L}=L$, $D(0)=DB$, and $\eta=0$.

We thank the referee for pointing this out and we fully agree. We have revised the text accordingly.

The discussion of Eq. (2) immediately following Eq. (2) is also hard to follow because the text states that z_0 and l are the only two fitting parameters, but kl appears in the equation. What happened to kl (it disappears in the diffusion limit in which $D(0)$ is DB)? Please alert the gentle reader.

We have dropped the explicit dependence of T from kl in Eq.(1) and Eq.(2) as well as the index $_{SC}$. This way the discussion should be clear.

For frequencies within the localization region, I'm not sure what is done. Is $D(0) = DB/(1-(kl)^2)$? If so, it would be helpful to explicitly say so here. If this is what is being done, then equation $D(0) = DB/(1-(kl)^2)$, introduced on page 5, should be set off and given an equation number so the reader more readily understands what is being done here. This is confusing because on page 5 it is stated that this equation is only valid when $kl \gg 1$, which is violated within the localization region.

We believe this remark still concerns the section 'Ballistic and diffuse transmission' (now renamed 'Total transmission coefficient in the multiple scattering regime'). In this section we do not consider any effects due to localization. With the changes made and explained above this should be clear in the revised version.

Moreover, we never use explicitly $D(0) = DB/(1-(kl)^2)$ in a fit since it is only valid for $(kl) \gg 1$ and therefore difficult to probe with our numerical data. We use the full results for $D(z)$ as provided by SC-theory. SC-theory, as implemented by us, however reproduces $D(0) = DB/(1-(kl)^2)$ in the appropriate limit, as expected.

Finally, I would like to see a plot of $D(z)$, since it plays such a key role in the analysis.

We have added a plot of $D(z)$ as Fig.3 in the section about the self-consistent theory.

Reviewer #2 (Remarks to the Author):

The authors have thoughtfully answered the referees' questions, and they have carefully and thoroughly revised the manuscript, including a new statistical analysis. It is a strong improvement that answers all of my essential questions.

We would like to thank the reviewer for the positive assessment.

Since I am no expert in the self-consistency theory, I instead focus in my review on the statistical analysis. Restricting the analysis to a least-square fit is a valid option, and I understand that a statistical hypothesis test is outside the scope of this paper. In that respect, the authors' answer in the rebuttal, which mentions the non-Gaussian statistics, is important and worth being included in the main text, for example, in an outlook to future research.

We have added the following sentence to the section 'self-consistent theory': *"In all cases, we perform a least-squared fit to $\ln T$ according to Eq.(2). We have considered other tests, such as χ^2 , but these other tests are often based on assumptions that are probably not met in our case. It is for example well known that $\ln T$ does not obey Gaussian statistics in the SAL regime (17)."*

The necessary approximations and the limitations of the complex fitting procedure make it difficult to assess whether the results are a proof or a strong indication of Anderson localization. In either case, the results deserve publication in Nature Communications, where my suggestion is to use a careful wording.

Minor questions and suggestions:

* On page 7: "that do not explicitly depend on ν ". Since the mobility edges are reduced frequencies, does this statement refer to the three different regimes that are studied?

Yes, the parameters we determine in the first step remain fixed throughout. We have carefully revised this section and hope the fitting procedure is clearer now.

* On page 9: Is there an intuitive interpretation of the "extrapolation length"? There is no definition when it first appears in the text.

We have removed this half sentence since the extrapolation length is properly introduced later in the text. We thank the referee for pointing this out.

* On page 10: The parentheses around k_l are missing in Eqs.~(1) and (2).

Following the advice of another referee we have removed (k_l) in these two equations since it doesn't enter the analysis at this point.

* On page 10: Which parameters in Eq.(1) do explicitly depend on k_l ?

We have removed k_l in Eq.(1) and have rephrased the discussion on page 11 to address this point.

* On page 10: The arguments of T_{SC} in Eq.(2) do not match the definition in Eq.(1).

We have corrected this.

* On page 10: "the number is kept constant at $N = 15$ ". What does this mean?

We have rephrased this sentence. We just want to be utmost clear how we calculate S using Eq.(2). We have included the exact values of L/a studied numerically as a table in the methods section.

* On pages 11 and 13: Please use a synonym of the word "significantly" to avoid confusion with its rigorous meaning in statistics.

We have removed "significantly" on page 11 and replaced it by "substantially" on page 13.

In what sense is the word used on pages 5 and 10?

We keep "significantly" on page 5. We have removed "significantly" on page 10 (now page 11).

* Page 29: Does the "proprietary software library" refer to MATLAB? Maybe it is worth emphasizing here that the main programs used here (MEEP and MPB) are open source software.

We have made this point clear on page 7 and under 'Code availability'.

* What is the estimator of the filling fraction? The volume fraction of the original 3D network, or the estimate of MEEP/MPB?

For the MPB calculations the filling fraction (ff) is 28.118% (resolution 128^3 voxels) and the $ff=28.119\%$ from the output of our own codes (set at 144^3 voxels). Both values are essentially the same with $10E-5$ precision. For MEEP, the ff is exactly the same within $1E-5$ % as well.

* What software was used for the Delaunay tessellation?

For the MPB calculations the Delaunay tessellation was performed with the Delaunay function from the `sicipy.spatial` open source library of Python. For the MEEP simulations we used an equivalent code programmed in MATLAB which will be uploaded to a repository (and made available before publication).

* Supplementary Material, page 5: Typo "for both and planes"

We have clarified this sentence.

* Supplementary Material, page 6: Typos "abut", "fr", Figure 2 (3 times)

We would like to thank the referee and we have corrected these typographical errors.

Reviewers' Comments:

Reviewer #1:

Remarks to the Author:

The authors have satisfied all my concerns in this revised manuscript and I recommend publication in Nature Communications. The authors present a very thorough numerical study of light transport in a class of hyperuniform disordered materials and show that the scaling theory of Anderson localization provides a better description of the numerical data than diffusion theory over a range of frequencies below (and to a lesser degree above) the band gap, consistent with theoretical arguments. This study will be of wide interest and significantly advances the field.

I do not need to see the manuscript again. Nevertheless, I have a couple of questions the authors may at their discretion want to address in the published version of their papers.

1. In the inset in Figure 2a, the two-parameter fit to the numerical data gives a reflection coefficient that appears to go to zero within the band gap, which is the opposite of what one expects physically (all light should be reflected). The authors may wish to comment on this in their manuscript.

2. I wonder what the implications of these numerical simulations are for experiments.

Reviewer #2:

Remarks to the Author:

The authors have answered all of my questions and further improved the presentation of their analysis and fascinating results.

Just a trifle remains: probably due to the changes of page numbers, the word "significantly" on former page 13, now page 18, was not replaced by "substantially" as the authors intended to do.

I am happy to recommend publication in Nature Communications.

Referee #1 writes: 1. In the inset in Figure 2a, the two-parameter fit to the numerical data gives a reflection coefficient that appears to go to zero within the band gap, which is the opposite of what one expects physically (all light should be reflected). The authors may wish to comment on this in their manuscript.

On page 10 we now write

...is linked to the angular averaged **specular** reflectivity R....

And we added: "We note that around $v' = a/\lambda_{\text{Gap}} \sim 0.45$ the fit is very poor and the fitted values of I and R become meaningless."

2. I wonder what the implications of these numerical simulations are for experiments.

On page 13 we have added:

"Moreover, our detailed results about the transition to SAL in realistic digital representations of dielectric networks can provide valuable guidance for future experimental attempts to probe light localization."

As suggested by referee #2 we have corrected 'significant' to 'substantially' on page 12.